# Altered Metabolism in Knockdown Lines of Two HXXXD/BAHD Acyltransferases During Wound Healing in Potato Tubers

**DOI:** 10.3390/plants13212995

**Published:** 2024-10-26

**Authors:** Jessica L. Sinka, Indira Queralta-Castillo, Lorena S. Yeung, Isabel Molina, Sangeeta Dhaubhadel, Mark A. Bernards

**Affiliations:** 1Department of Biology, Western University, London, ON N6A 5B7, Canada; jsinka2@uwo.ca (J.L.S.); indiraqueralta@yahoo.com (I.Q.-C.); syeung96@uwo.ca (L.S.Y.); sangeeta.dhaubhadel@agr.gc.ca (S.D.); 2Department of Biology, Algoma University, Sault Ste. Marie, ON P6A 2G4, Canada; isabel.molina@algomau.ca; 3London Research and Development Centre, Agriculture and Agri-Food Canada, 1391 Sandford St., London, ON N5V 4T3, Canada

**Keywords:** potato, *Solanum tuberosum*, suberin, wound healing, HXXXD/BAHD acyltransferase

## Abstract

Suberin biosynthesis involves the coordinated regulation of both phenolic and aliphatic metabolisms. HXXXD/BAHD acyltransferases occupy a unique place in suberization, as they function to crosslink phenolic and aliphatic monomers during suberin assembly. To date, only one suberin-associated HXXXD/BAHD acyltransferase, StFHT, has been described in potatoes, whereas, in *Arabidopsis*, at least two are implicated in suberin biosynthesis. RNAseq data from wound-induced potato tubers undergoing suberization indicate that transcripts for 28 HXXXD/BAHD acyltransferase genes accumulate in response to wounding. In the present study, we generated RNAi knockdown lines for *StFHT* and another highly wound-induced HXXXD/BAHD acyltransferase, designated *StHCT*, and characterized their wound-induced suberin phenotype. StFHT-RNAi and StHCT-RNAi knockdown lines share the same aliphatic suberin phenotype of reduced esterified ferulic acid and ferulates, which is similar to the previously described StFHT-RNAi knockdown suberin phenotype. However, the phenolic suberin phenotype differed between the two knockdown genotypes, with StHCT-RNAi knockdown lines having proportionately more *p*-hydroxyphenyl-derived moieties than either StFHT-RNAi knockdown or empty vector control lines. Analysis of soluble polar metabolites revealed that StHCT catalyzes a step upstream from StFHT. Overall, our data support the involvement of more than one HXXXD/BAHD acyltransferase in potato suberin biosynthesis.

## 1. Introduction

The tissue-specific modification of plant cell walls facilitates diverse functions throughout growth and development and in response to stress. Three major cell wall modifications involve the deposition of high molecular weight polymers, including lignin, cutin, and suberin. Lignin, deposited within the cell walls of vascular tissues (vessels, trachieds, and fibers), is a monolignol-based polymer, crosslinked through C-C and ether bonds, and provides structural support [1]. Cutin is an aliphatic poly(ester) comprised largely of fatty acids and hydroxylated fatty acids embedded within the cell walls of above-ground epidermal tissues [2]. Cuticular waxes (fatty acids, 1-alkanols, alkanes, aldehydes, and ketones), embedded within the cutin polymer and on the external surface, provide a nearly perfect barrier to non-stomatal water loss [2]. By contrast, suberized cells have a poly(phenolic) polymer embedded within their cell walls, to which an aliphatic-based poly(ester) is attached on the inner surface of the wall (i.e., between the cell wall and plasma membrane) [3]. Suberized tissue provides protection against pathogen infection and water loss [3], making it a target for crop improvement (i.e., enhanced pathogen resistance and storage longevity).

The aliphatic-based poly(ester) of suberin is comprised of fatty acids, 1-alkanols, ω-hydroxy fatty acids, α,ω-dioic acids, hydroxycinnamic acids, and glycerol [3]. Like cutin, the aliphatic-based component of the suberin poly(ester) has embedded waxes including fatty acids, 1-alkanols, alkanes, and hydroxycinnamoyl esters of 1-alkanols (i.e., ferulates) [3] that provide protection against water loss in belowground epidermal tissues (root epidermis, endodermis, and tuber periderm) and the bark of trees and shrubs. One unique aspect of suberin biosynthesis and assembly is the covalent linking of phenolic and aliphatic components. These linkages are found within the poly(ester) (i.e., esterified hydroxycinnamic acids) and associated waxes (i.e., ferulates) and at the interface between the poly(phenolic) and poly(ester) components. Conjugations between hydroxycinnamic acids and aliphatics (esp. 1-alkanols, ω-hydroxy fatty acids, and glycerol) are catalyzed by **B**enzyl alcohol *O*-acetyltransferase-**A**nthocyanin *O*-hydroxycinnamoyltransferase-N-**H**ydroxycinnamoyl/benzoyltransfease-**D**eacetylvindoline 4-*O*-acetyltransferase (BAHD) family acyltransferases, several of which have been characterized in suberizing tissue of *Arabidopsis* [4,5,6], poplar [7], and potato [8,9,10]. Enzymes in the BAHD acyltransferase superfamily are characterized by a HXXXD consensus motif [11] and catalyze the transfer of acyl moieties from a range of coenzyme A-thioester donors (e.g., hydroxycinnamoyl-CoAs, acetyl-CoA, malonyl-CoA, and benzoyl-CoA) to acceptor alcohols [11,12].

Two HXXXD-motif family acyltransferases in *Arabidopsis* aliphatic suberin feruloyl transferase (ASFT; *At5g41040*) [4,5] and fatty alcohol:caffeoyl-CoA caffeoyl transferase (FACT; *At5g63560*) [6] function in the biosynthesis of the aliphatic suberin polyester and wax-associated hydroxycinnamoyl esters. *Atasft* loss-of-function T-DNA insertion mutants show a significant reduction in esterified ferulate content in both root and seed coat suberin [4,5]. Alkyl hydroxycinnamate esters in root-associated waxes, however, were not affected in the *Atasft* knockout mutants [5]. Recombinant AtASFT functions as a feruloyl transferase by catalyzing the conjugation between ferulic acid and ω-hydroxy fatty acids and fatty alcohols. In vitro enzyme analyses also support the transfer of caffeoyl-CoA and *p*-coumaroyl-CoA to fatty alcohols and fatty acids groups by AtASFT [4,5,13]. In contrast, *Atfact* knockout mutants show a significant decrease in alkyl caffeates, a small reduction in alkyl ferulates, and a significant increase in alkyl coumarates, with no changes in root suberin polyester composition. Instead, the caffeate content is reduced in seed coat suberin [6]. In vitro enzyme assays confirmed *in planta* analyses, showing that recombinant AtFACT catalyzes the transfer of caffeic acid from caffeoyl-CoA to fatty alcohols. Recombinant AtFACT can also use feruloyl-CoA and coumaroyl-CoA as acyl donors in vitro [6,13]. The unique suberin profiles of *Atasft* and *Atfact* mutants suggest multiple HXXXD/BAHD acyl transferases are required for complete suberin biosynthesis. Furthermore, the covalent linkage between phenolic and aliphatic domains of suberin in potato is hypothesized to be a glycerol-bridged hydroxycinnamic acid-acyl ester [14], though a direct hydroxycinnamic acid-acyl ester cannot be ruled out [15]. Either case invokes a role for additional HXXXD/BAHD acyltransferases in suberin formation.

Homologs of *AtASFT* have been described for poplar [7] and potato [8]. The potato homolog *Fatty ω-hydroxyacid*/*fatty alcohol hydroxycinnamoyl transferase* (*StFHT*; *Soltu.DM.03G020160.1*) has been characterized [8,9,10]. *StFHT* RNAi native periderm has a reduced content of esterified ferulic acid, ω-hydroxyacid (C18:1), and primary alcohols [8]. The alkyl ferulate and alkane levels were also reduced in the suberin-associated waxes [8]. Solid-state ^13^C nuclear magnetic resonance (NMR) analysis revealed more flexible aliphatic chains and substantial amounts of poly(aromatics) [9]. Alkyl ferulates were also reduced in the *StFHT* RNAi wound periderm, while there were no statistical differences in the amounts of guaiacyl- and syringyl-derived thioacidolysis products in the suberin phenolic domain between the wildtype (WT) and the FHT knockdown plants, and no *p*-hydroxyphenyl units were found [10].

Wounding of potato tubers induces the expression of 28 putative HXXXD/BAHD acyltransferases [16]. While transcript accumulation for *StFHT* (*Soltu.DM.03G020160.1*) was highest, another putative hydroxycinnamoyl acyltransferase (HCT) gene (Soltu.DM.03G031830.1; *StHCT*) with high homology to *AtASFT* and *AtFACT* was also strongly induced by wounding. To facilitate a better understanding of suberization, we generated RNAi knockdown lines for both *StHCT* and *StFHT*. In the present work, a comparative analysis of the impacts of StFHT and StHCT knockdown was performed with regards to the composition of both native and wound periderms and both wound-induced metabolism and wound suberin formation. Our data provide support for the role of the poly(phenolic) component of suberin in its function as a barrier to water loss and identify potential new targets for improving tuber storage.

## 2. Results

### 2.1. Identification of Candidate Genes

The initial identification of novel putative HXXXD/BAHD genes associated with suberization [17] was based on the DM v4.03 potato genome [18]. To select candidate genes, a BLASTp query using protein sequences of previously characterized potato feruloyl hydroxycinnamoyl transferase (StFHT; PGSC0003DMG400031731 = Soltu.DM.03G020160.1), two *Arabidopsis* HXXXD/BAHD genes: aliphatic suberin feruloyl transferase (*AtASFT*; *At5g41040*) [4,5] and *fatty alcohol:caffeoyl-CoA caffeoyl transferase* (*AtFACT*; *At5g63560*) [6], and a novel HXXXD/BAHD gene (*At1g24430*) that co-expresses with *ASFT* and *FACT* in suberizing *Arabidopsis* tissues [19,20] was conducted. Refinement of the search results, based on sequence similarity (>50%), low E-value, and high expression levels post-wounding [16], yielded 11 candidates. The phylogenetic relationship between the 11 potato and three *Arabidopsis* HXXXD/BAHD gene products, based on their amino acid sequences, revealed tight clustering between StFHT, AtASFT, and AtFACT (data not shown). A closely related pair of proteins included an uncharacterized “Hydroxycinnamoyl Transferase” (StHCT) that had high transcript accumulation, second only to StFHT. The gene for this protein (PGSC0003DMG400014152 = Soltu.DM.03G031830.1) was chosen for further characterization. Similarly, the closest related protein to At1g24430 that showed wound-induced transcript accumulation (PGSC0003DMG400018015 = Soltu.DM.05G020490.1; herein *St430*) was chosen for further characterization.

A subsequent, unbiased analysis of the revised potato genome (DM v6.1) [21] revealed 160 genes annotated as HXXXD/BAHD acyltransferases. Redundant entries and short/truncated proteins were first removed and the remaining sequences cross-referenced to the potato genome (DM v4.03) for which we have wound-induced expression data [16]. A further 22 sequences were identified as either duplicates (i.e., more than one DM v6.1 amino acid sequence matched the same DM v4.03 sequence) or were a low match to the DM v4.03 sequences. Of the remaining 75 genes, 28 were induced by wounding [16]; the predicted amino acid sequences for wound-induced genes were selected to create a potato HXXXD/BAHD acyltransferase phylogenetic tree (Figure 1) (See Appendix A for a complete phylogenetic tree encompassing all 75 potato HXXXD/BAHD acyltransferases). StFHT (Soltu.DM.03G020160.1) clustered tightly with At5g41040 and At5g63560, whereas StHCT (Soltu.DM.03G031830.1), annotated as a Hydroxycinnamoyl-CoA Shikimate/Quinate Hydroxycinnamoyl Transferase (HS/QT) in DM v6.1, formed a nearest-neighbor subcluster along with the same HXXXD/BAHD gene as before, also annotated as a HS/QT (Soltu.DM.07G000810.1) (Figure 1). Three additional genes annotated as HS/QTs formed a separate distant subcluster. St430 (Soltu.DM.05G020490.1) formed a small distant subcluster with At1g24430 (Figure 1).

### 2.2. Selection of RNAi Knockdown Lines

Independent knockdown lines were generated for *StFHT*, *StHCT,* and *St430*. Empty vector (EV) control lines were also generated. Microtubers obtained for each independent transgenic line were wounded and incubated for 72 h. Suberizing tissue was collected and RNA isolated for qRT-PCR (Figure 2). While multiple independent knockdown lines were generated for both *StFHT* (Figure 2C) and *StHCT* (Figure 2D), only one *Soltu.DM.05G020490.1* line with reduced gene expression was obtained (Figure 2B). Three lines each of StFHT-RNAi and StHCT-RNAi, showing <50% of WT transcript accumulation, were selected for further characterization, along with the single St430-RNAi line. Since the expression levels for *StFHT*, *StHCT,* and *St430* did not differ between the WT and EV lines (Figure 2A), EV lines were used as controls for all subsequent analyses.

### 2.3. StHCT and StFHT RNAi Lines Share a Common Aliphatic Suberin Phenotype

The aliphatic suberin, i.e., poly(aliphatic), profiles of our three independent StFHT- and StHCT-RNAi lines were not significantly different (Appendix A); consequently, we pooled the data within each genotype. The aliphatic suberin phenotype of our single St430-RNAi line did not differ from the EV lines (Figure 3 and Appendix A). However, our independently generated StFHT- and StHCT-RNAi lines had a similar aliphatic suberin phenotype to that described by Serra et al. (2010) [8] for their StFHT RNAi lines, most notably a significant reduction in the amount of esterified ferulic acid (Figure 3 and Appendix A). Our StFHT-RNAi lines further showed a slight reduction in ω-hydroxy fatty acids (Figure 3). However, we did not observe any significant differences in the amounts of the other common suberin aliphatic monomers between EV and any of our RNAi lines (Figure 3 and Appendix A).

The StFHT-RNAi lines described by Serra et al. 2010 [8] also had reduced levels of soluble alkyl ferulates. In our hands, all RNAi lines, including St430-, StFHT-, and StHCT-RNAi, had reduced alkyl ferulate levels relative to the EV controls (Figure 4 and Appendix A).

The similarity in aliphatic suberin phenotype between the StHCT and StFHT RNAi lines raised the possibility of off target RNAi silencing. However, *StHCT* gene expression in the StFHT RNAi lines was unaffected and vice versa (Appendix A).

### 2.4. StHCT and StFHT RNAi Lines Have Different Phenolic Suberin Phenotypes

The preliminary phenolic suberin analysis of our St430-RNAi line yielded a phenolic profile identical to that of EV lines (data not shown). Given the lack of difference in suberin composition (both aliphatic and phenolic) between the St430-RNAi and EV lines, further analysis focused on EV and StHCT- and StFHT-RNAi lines.

Microscale nitrobenzene oxidation (mNBO) was used to assess the phenolic composition of 7-day wound-healed periderm and native periderm tissues from EV and StFHT- and StHCT-RNAi knockdown lines. The main products from oxidative cleavage of the α-β bonds of poly(phenolics) are largely aromatic aldehydes and their associated carboxylic acids [22]. To assess the total phenolic suberin, the TMS derivatives of six major NBO products were quantified, including 4-hydroxybenzaldehyde and 4-hydroxybenzoic acid (*p*-hydroxyphenyl-derived), vanillin and vanillic acid (guaicyl-derived), and syringaldehyde and syringic acid (syringyl-derived).

The native periderm of potato comprises *p*-hydroxyphentyl-, guaiacyl-, and syringyl-derived moieties, with guaiacyl moieties predominating (Figure 5). The total amount of phenolic suberin did not differ between EV and StFHT-RNAi lines (Figure 6); however, StHCT-RNAi knockdown lines produced significantly less phenolic suberin than either EV or StFHT-RNAi knockdown lines. The lower amount of total phenolic suberin in native StHCT-RNAi line periderm was reflected in lower amounts of all three types of phenolics (Appendix A). For example, StHCT-RNAi native periderm was significantly depleted in G- and S-derived products in comparison to EV and StFHT-RNAi (*p* < 0.05) lines. StHCT-RNAi also had significantly less P-derived products than StFHT-RNAi (*p* < 0.01), but neither RNAi line significantly differed from the EV (Appendix A).

Both StFHT-RNAi and StHCT-RNAi lines produced more wound periderm phenolic suberin than their respective native periderms, while EV produced equal amounts of wound and native periderm phenolic suberin (Figure 6). However, the wound periderm phenolic composition of the StHCT-RNAi lines differed significantly from its native periderm, while that of the StFHT-RNAi lines and the EV did not. Specifically, knocking down HCT resulted in a wound periderm with equal amounts of *p*-hydroxyphenyl and guaiacyl moieties (Figure 5F). The latter was the result of increased *p*-hydroxyphenyl moiety incorporation in the wound periderm, as the amount of guaiacyl moieties was similar in both wound and native periderm.

### 2.5. Time Course Analysis Reveals an Early Role for StHCT and StFHT During Wound Healing

In response to wounding, potato tubers undergo a rapid shift in metabolism over the course of two days before establishing a new metabolic steady state. This is reflected in PLS-DA plots of polar metabolite profiles derived from EV tubers (Figure 7A), which form distinct clusters for 0, 12, 24, and 48 h post-wounding (hpw). Here, PC1, representing 73% of the variance between profiles, accounts for a major shift between 0 and 12 hpw, while PC2 (7%) accounts for the separation between profiles from 12, 24, and 48 hpw. In contrast, metabolite profiles from StFHT-RNAi lines (Figure 7B) separate along PC2 (16%) between 0, 12, and 24 hpw before showing strong separation along PC1 (59%). A similar plot was obtained for StHCT-RNAi lines (Figure 7C), except that the initial separation between metabolite profiles from 0, 12, and 24 hpw along PC2 (24%) is more exaggerated; PC1 (57%), however, accounts for the largest separation leading to a distinct metabolite profile at 48 hpw.

### 2.6. Targeted Metabolite Analysis Reveals a Potential Role for StHCT Early in Phenylpropanoid Metabolism

The reduction in esterified ferulic acid and ferulates in the suberin of the StHCT- and StFHT-RNAi lines and suberin phenolics in the StHCT-RNA lines raised the question “What is the metabolic fate of hydroxycinnamic acids normally destined for the suberin polymer”? To address this question, we focused our attention on changes in common hydroxycinnamic acid derivatives, including glycosides, amides, and esters, during the early stages (i.e., first two days) of wound healing. The most abundant soluble hydroxycinnamoyl-conjugates found were putrescine and tyramine amides and chlorogenic acid (Figure 8), with only trace amounts of glycosides. At 12 hpw, StFHT-RNAi lines accumulated caffeoyl and *p*-coumaroyl conjugates in greater abundance than EV lines, specifically *p*-coumaroyl tyramine (*p* = 0.0930; Figure 8A), chlorogenic acid (*p* < 0.05; Figure 8C), and caffeoyl putrescine (*p* < 0.0001; Figure 8D). At 24 hpw, the caffeoyl and *p*-coumaroyl conjugate pools in the StFHT-RNAi lines were no longer statistically different from those of the EV lines; however, the StHCT-RNAi lines had accumulated significantly more *p*-coumaroyl tyramine (*p* < 0.05; Figure 8A), caffeic acid (*p* < 0.01; Figure 8B), chlorogenic acid (*p* < 0.05; Figure 8C), and caffeoyl putrescine (*p* < 0001; Figure 8D) than the EV lines. In contrast, despite an apparent spike at 0 hpw, the StHCT-RNAi lines were depleted in feruloyl putrescine at 12 (*p* < 0.01), 24 (*p* < 0.05), and 48 hpw (*p* < 0.05) (Figure 8E). While feruloyl putrescine was significantly lower in StFHT-RNAi lines relative to EV lines (*p* < 0.001) at 12 hpw, there was no significant difference in feruloyl putrescine accumulation between St-FHT-RNAi lines and EV lines at later time points (Figure 8E). Similarly, feruloyl tyramine was significantly elevated in StHCT-RNAi lines, relative to that of EV at 12 hpw (*p* < 0.05), while the StFHT-RNAi line feruloyl tyramine levels were not statistically different from EV (*p* = 0.9963). However, 12 h later, at 24 hpw, the StFHT-RNAi lines accumulated significantly greater amounts of feruloyl tyramine than EV (*p* < 0.01) and STHCT-RNAi lines (Figure 8F). Overall, StHCT-RNAi lines accumulated more *p*-coumaroyl putrescine and tyramine amides than either EV or StFHT-RNAi lines, especially at early time points post-wounding.

### 2.7. StHCT-RNAi Affects the Permability of Native Periderm but Not Wound Periderm

Wound periderm was more permeable than native periderm for all genotypes (Figure 9) by a factor of nearly two. There was no statistical significance in the rate of water loss from wound periderm from any genotype. By contrast, within the native periderm samples, water loss was significantly greater for StHCT-RNAi lines (*p* < 0.05), while native periderm from EV and StFHT-RNAi lines were equal (Figure 9).

## 3. Discussion

The assembly of the suberin polymer requires the coordination of both phenolic and lipid metabolism [3]. One working hypothesis [3] posits that the poly(phenolic) domain is laid down first, followed by the poly(aliphatic) domain. It follows that crosslinking between the domains involves covalent linkages between phenolic and aliphatic monomers. One group of enzymes that can transfer hydroxycinnamoyl moieties to acceptor aliphatics are the HXXXD/BAHD family acyl transferases [11]. To date, two BAHD family HXXXD-type acyltransferases in *Arabidopsis* (aliphatic suberin feruloyl transferase (At*ASFT*; *At5g41040*) [4,5] and fatty alcohol:caffeoyl-CoA caffeoyl transferase (At*FACT*; *At5g63560*) [6]), and At*ASFT* homologs in poplar [7] and potatoes [8] have been characterized. While their function as acyl transferases responsible for soluble alkyl-hydroxycinnamates (principally, alkylferulates) is clear, a role for them in linking the poly(phenolic) and poly(aliphatic) domains of suberin remains to be determined. However, recent data [15] strongly support this role for acyl transferases.

In potato, there are 160 genes annotated as HXXXD/BAHD acyl transferases, of which, 75 are full length and non-redundant, and 28 are expressed in response to wounding [16]. Amongst these is *StFHT*, the potato homolog of At*ASFT*. The principal suberin phenotype of StFHT RNAi native periderm is a reduced content of esterified ferulic acid, ω-hydroxyacid (C18:1), primary alcohols, and alkyl ferulates, while wound suberin is characterized by reduced levels of alkyl ferulates [8]. Since the total amounts of poly(phenolic) and poly(aliphatic) components of suberin were largely unaffected by StFHT RNAi knockdown [8], we reasoned that additional acyltransferases are involved and looked to wound-induced HXXXD/BAHD acyl transferases for candidates.

Transcripts of Soltu.DM.03G031830.1 are the second-most abundant of any HXXXD/BAHD acyl transferase (transcripts of St*FHT* are highest) post-wounding [16]. Our initial selection of this gene was, in part, based on an earlier annotation of the potato genome (DM v4.03) [18], in which it was annotated as a hydroxycinnamoyl acyl transferase (St*HCT*), and its close phylogenetic relationship with St*FHT*. More recently. Soltu.DM.03G031830.1 was re-annotated as a hydroxycinnamoyl-quinate/shikimate hydroxycinnamoyl transferase (St*HQ*/*ST*) [21]. In our phylogeny of the 75 non-redundant HXXXD/BAHD acyl transferases in potatoes, StHCT clustered with another StHQ/ST protein, albeit separately from a second cluster of three others also annotated as StHQ/ST (Figure 1 and Appendix A). Transcripts for all five annotated St*HQ*/*STs,* along with St*HCT* and St*FHT,* accumulate post-wounding [16]. So too do transcripts of *Soltu.DM.05G020490.1* (designated herein as *St430* due to its close phylogenetic relationship to *At1g24430*, an *Arabidopsis* acyltransferase co-expressed with At*ASFT* and At*FACT* in suberizing tissues).

### 3.1. StHCT and StFHT Function Non-Redundantly During Wound-Induced Suberization

While the St430-, StFHT-, and StHCT-RNAi lines showed reduced levels of alkyl ferulates, only the StFHT- and StHCT-RNAi lines showed reduced levels of esterified ferulic acid. suggesting a similar function. The knockdown of St430 had no other impact on either aliphatic or phenolic suberin, suggesting a minor role (if any) for this gene in potato suberin formation. In contrast, the different composition of wound-induced phenolic suberin between the StFHT- and StHCT-RNAi lines, with a higher proportion of *p*-hydroxyphenyl moieties and reduced guaiacyl moieties in the StHCT-RNAi lines, argues that the observed differences in aliphatic suberin phenotype between knockdown lines of these two genes arises from two different routes. In the case of StFHT-RNAi lines, previous characterization clearly demonstrates that StFHT functions to transfer ferulic acid to 1-alkanols (to yield alkyl ferulates) and ω-hydroxyacids [8,9,10], ultimately yielding the esterified ferulic acid typical of potato aliphatic suberin. In the case of StHCT-RNAi lines, the reduction in esterified ferulic acid and ferulates likely arise from a reduced pool of available ferulic acid. This conclusion is supported by the accumulation of higher amounts of soluble *p*-coumaroyl and caffeoyl derivatives during wound healing and the disproportionately high levels of *p*-hydroxyphenyl moieties in the wound phenolic suberin of StHCT-RNAi line, compared to either StFHT-RNAi or EV lines.

### 3.2. A Role for StHCT Upstream of StFHT

In phenylpropanoid metabolism, *p*-coumaroyl-CoA is hydroxylated to form caffeoyl-CoA, which is then methylated to yield feruloyl-CoA (Figure 10) (see, for example, [23]). However, there are two routes to caffeoyl-CoA: direct hydroxylation by *p*-coumarate-3-hydroxylase and indirect hydroxylation by *p*-coumaroyl quinate-3′-hydroxylase [23]. The key step in the indirect hydroxylation of *p*-coumaroyl-CoA is its conversion to *p*-coumaroyl quinate (or shikimate) by the enzyme hydroxycinnamoyl quinate/shikimate transferase. After hydroxylation of *p*-coumaroyl quinate to caffeoyl quinate (i.e., chlorogenic acid), hydroxycinnamoyl quinate/shikimate transferase transfers the caffeoyl moiety back to CoA [23].

The wound suberin chemotype of StHCT-RNAi lines supports an in vivo role for St*HCT* as a hydroxycinnamoyl quinate/shikimate transferase, consistent with its recent annotation [21]. The disproportionately higher amount of *p*-hydroxyphenyl moieties in StHCT-RNAi line wound phenolic suberin likely results from a slower production of ferulic acid-derived guaiacyl moieties under reduced St*HCT* expression during the wound response. That a significant amount of caffeoyl derivatives, but especially chlorogenic acid, also accumulate in StHCT-RNAi lines reflects the dual role of hydroxycinnamoyl quinate/shikimate transferases in general phenylpropanoid metabolism. A significant amount of ferulic acid and its derivatives still accumulate in StHCT-RNAi lines, reflecting the incomplete knockdown of St*HCT* and likely contribution of the direct hydroxylation of *p*-coumaroyl-CoA by *p*-coumarate-3-hydroxylase; the gene for *p*-coumarate-3-hydroxylase (St*pC3H*) is induced by wounding [23], albeit at a lower level than St*HCT*.

In contrast, the suberin chemotype of StHCT-RNAi knockdown line native periderm closely resembles that of EV lines, albeit in a lower total amount. However, it is not clear whether St*HCT* expression is reduced in non-wounded tissue of StHCT-RNAi lines. Regardless, the slower developmental deposition of native periderm and likely contribution by St*p*C3H provides potential explanation.

### 3.3. Phenolic Suberin Composition, but Not Amount, Impacts Suberin Permeability

The wound periderms tissue of our EV, StFHT-RNAi, and StHCT-RNAi lines are more permeable than their native periderm counterparts, consistent with previous reports (e.g., [9]). All three genotypes produced similar wound phenolic and aliphatic suberin in roughly equal amounts. It was somewhat surprising, therefore, that, in the case of the StFHT-RNAi and StHCT-RNAi lines, higher amounts of wound periderm phenolics were found than in their native periderms. However, the greater permeability of wound periderm likely reflects the incomplete nature of the closing layer [24] rather than the total amount of phenolic and/or aliphatic suberin.

In contrast, the permeability of native periderm tissue, which is laid down over a longer timeframe and can involve more cell layers than early-stage wound periderm [24], differed across the EV, StFHT-RNAi, and StHCT-RNAi lines. Specifically, StHCT-RNAi line native periderm had the highest permeability, followed by EV and StFHT-RNAi native periderms. The native periderm of these lines, but especially StHCT-RNAi, differed compositionally. StHCT-RNAi phenolic suberin had proportionally higher *p*-hydroxyphenyl-derived monomers and lower guaiacyl-derived monomers. These data suggest that the phenolic composition of potato suberin contributes to its water retention function, presumably by creating a hydrophobic barrier within the cellulosic walls between adjacent periderm cells lined with aliphatic-derived lamellae (between the cell wall and plasma membrane).

## 4. Conclusions

Suberization is a highly coordinated process involving both phenolic and aliphatic metabolism. HXXXD/BAHD acyltransferases are integral to the overall assembly of suberin, and in potatoes, St*FHT* has been shown to function in the process [8]. However, in *Arabidopsis*, at least two HXXXD/BAHD acyltransferases are involved in suberin biosynthesis [4,5]. In the present study, we sought to identify additional HXXXD/BAHD acyltransferases involved in potato suberin biosynthesis. We generated RNAi knockdown lines for *StFHT* and another highly wound-induced HXXXD/BAHD acyltransferase, designated *StHCT*. StFHT-RNAi and StHCT-RNAi knockdown lines shared the same aliphatic suberin phenotype of reduced esterified ferulic acid and ferulates, which was similar to the previously described StFHT-RNAi knockdown suberin phenotype [8]. However, the phenolic suberin phenotype differed between the two knockdown genotypes, with the StHCT-RNAi knockdown lines having proportionately more *p*-hydroxyphenyl-derived moieties than either StFHT-RNAi knockdown or empty vector control lines. The analysis of soluble polar metabolites revealed that StHCT catalyzes a step upstream from StFHT. Overall, our data support the involvement of more than one HXXXD/BAHD acyltransferase in suberin biosynthesis, with evidence supporting the incorporation of phenolics biosynthesized de novo in response to wounding

## 5. Materials and Methods

### 5.1. Chemicals and Vectors

All solvents and reagents were purchased from Thermo Fisher Scientific and Sigma-Aldrich unless otherwise specified. The Gateway^®^ BP Clonase^™^ enzyme mix, the LR Clonase^™^ enzyme mix, and the entry vector (pDONR^™^/Zeo), were purchased from Invitrogen. The silencing (destination) vector pK7GWIWG2D(II),0 [25] was obtained from VIB-UGent Center for Plant Systems Biology (https://gatewayvectors.vib.be/).

### 5.2. Plant Material and Growth Conditions

Commercial potato (*S. tuberosum*) cultivar Desiree was obtained as in vitro plantlets from the Canadian Potato Variety Repository at the Plant Propagation Center (New Brunswick) (https://www2.snb.ca/content/snb/en/services). Plantlets were propagated and maintained in Murashige and Skoog (MS) basal salt medium (pH 5.8) supplemented with 3% (*w*/*v*) sucrose and solidified with 0.22% gelrite^®^. Plantlets were grown in a growth cabinet under a light/dark photoperiod cycle of 16/8 h at 22 °C and 67 µmol m^−2^ s^−1^ [26,27]. Nodal sections containing at least one leaf were excised and subcultured into fresh medium every 4–6 weeks. For in vitro microtuber formation, nodal sections (0.5–1.0 cm long) containing at least one leaf were cut from axenic plantlets, transferred to MS medium supplemented with 8% (*w*/*v*) sucrose, and incubated for 1 week under a short-day photoperiod (8/16 h light/dark) at 22 °C and 67 µmol m^−2^ s^−1^ before transfer into continuous dark conditions at 22 °C [28]. Minitubers were generated by transferring in vitro plantlets to soil (Promix BX^®^) in 4″ round pots for hardening. Plants were watered as needed and mature tubers harvested after ZZ days.

### 5.3. Identification of Potato Homologs of A. thaliana Candidate Genes

Full-length sequences of *Arabidopsis* genes *At1g24430*, ASFT (*At5g41040*), and FACT (*At5g63560*) were obtained from The *Arabidopsis* Information Resource (TAIR) database (https://www.arabidopsis.org/index.jsp). Potato homologs were identified through the search tools at the Spud DB Potato Genomics Resource (http://solanaceae.plantbiology.msu.edu/index.shtml) based on the DM v4.03 potato genome. Potato wound response RNA transcriptome data [16] were used to narrow down potato candidates with potential acyltransferase function to those involved in wound suberin formation. The sequence ID search tool from the Spud DB Potato Genomics Resource was used to identify *A. thaliana* protein matches based on percent similarity (>50%) and a lower expected value (E-value) (closer to zero for significant match). The RNA transcriptome data were used again to narrow down the potato candidates lists (5–10), using their temporal expression patterns post-wounding. Inferred protein sequences of the *Arabidopsis* genes and the potato candidates were retrieved in FASTA format for multiple sequence alignment. Amino acid sequence alignment was performed using the MUSCLE program in MEGA version 6 [29]. A phylogenetic tree was constructed using a neighbor-joining algorithm with 1000 bootstrap trials. Multiple sequence alignment using the nucleic acids of potato candidate genes were analyzed and edited in DNAMAN to identify conserved regions or areas with high similarity for RNA interference (RNAi) construct design (Appendix A).

### 5.4. Potato HXXXD/BAHD Phylogeny

The revised potato genome (DM v6.1; [21]) was searched using HXXXD/BAHD acyltransferase as a keyword, yielding 160 unique sequences. The amino acid sequence files were retrieved and processed. Redundant entries and short/truncated proteins were first removed and the remaining sequences cross-referenced to the potato genome (DM v4.03). Duplicate and low match sequences were removed. The remaining 75 genes and, separately, the 28 induced by wounding [16] were used to create a potato HXXXD/BAHD acyltransferase neighbor-joining phylogenetic tree (1000 bootstraps) using MAFFT (mafft.cbrc.jp) based on amino acid sequences. Amino acid sequences for three *Arabidopsis* HXXXD/BAHD genes (*At5g41040*, *At5g63560,* and *At1g24430*) were also included. StCYP86A33 (Soltu.DM.07G002220.1) was used as an outlier to root the tree. The rooted phylogenetic tree was rendered using Phylo.IO (http://phylo.io/ [30])

### 5.5. RNAi Plasmid Construction

Gene-specific RNAi silencing constructs were generated using Gateway^®^ cloning (Invitrogen). Gene-specific fragments (322–351 bps) were PCR-amplified from cDNA derived from messenger RNA (mRNA) isolated from wound-healing potato tubers (3 dpw) using gene-specific primers (Appendix A) that included the *attB1* (for the forward primers) and *attB2* (for the reserve primers) Gateway^®^ recombinant sequences, respectively, at their 5′ ends. The PCR conditions were as follows: 94 °C for 4 min, 30 cycles of 94 °C for 30 s, 59 °C for 1.30 min, 72 °C for 1.30 min, and a final extension time at 72 °C for 7 min. After PCR amplification, the products were separated on 1.5% agarose gel and a single band for each gene-specific amplicon excised from the gel and purified using the QIAquick Gel Extraction Kit (Qiagen, Hilden, Germany), following the manufacturer’s protocol. The purified PCR products were cloned into the gateway entry vector pDONR/Zeo using the BP clonase II mix (Invitrogen) following the manufacturer’s protocol, and the reaction products were transferred into DH5α competent *E. coli* cells by the 90 s heat shock method [31]. The DH5α *E. coli* were transferred to low salt agar Luria–Bertani (LB) medium plates containing 100 µg/mL zeocin and incubated overnight at 37 °C. Positive colonies (confirmed by colony PCR) were picked, transferred to LB liquid broth, and incubated overnight with shaking at 37 °C. The overnight culture was then used to extract the plasmid DNA using the QIAprep Spin Miniprep Kit (Qiagen), following the manufacturer’s protocol. After PCR confirmation (as above), the pDONR-*StFHT*, *StHCT*, and *Soltu.DM.05G020490.1* plasmids were cloned into the RNAi destination vector pK7GWIWG2D(II) [25] using the LR clonase II mix (Invitrogen^®^), according to the manufacturer’s instructions. The LR reaction products were transferred into DH5α competent *E. coli* cells by the heat shock method as above and grown on LB plates containing 100 µg/mL spectinomycin. To confirm that the final constructs had the gene-specific inserts in the correct orientation to create the hairpin RNA loops, bacteria colonies were screened by PCR using gene-specific forward primers designed for the genotype analysis and the destination vector intron (the chloramphenicol resistance, Cmr, area) forward (CmrF: 5′-CGA TTC AGG TTC ATC ATG CCG TCT-3′) and reverse (CmrR: 5′-TGA GCA ACT GAC TGA AAT GCC TCC-3′) primers (Appendix A). Positive colonies were picked and grown in LB liquid medium for a bulk extraction of plasmid DNA. The sequences of the final constructs were verified by Sanger sequencing at the London Regional Genomics Center (LRGC).

### 5.6. Plant Transformation for RNAi-Mediated Gene Silencing

The RNAi recombinant plasmids (pK7GWIWG2D(II)RNAi) were transferred into *Agrobacterium tumefaciens* strain GV2260 [32] and used to transform potato internodal explants [33,34]. Briefly, the RNAi final constructs (pK7GWIWG2D(II)-*StFHT*, *StHCT*, and *Soltu.DM.05G020490.1*) were transferred into the *Agrobacterium strain* GV2260 by electroporation using the MicroPulser Electroporator (Bio-Rad). For the electroporation steps, 0.5–1 µg of plasmid DNA (5 µL) was added into the *Agrobacterium* competent cells (50 µL/tube) and kept on ice for 15 min. Each individual competent cell/plasmid mixture was transferred to a prechilled cuvette (0.1 cm gap) and pulsed once for 5 milliseconds under the Agr mnemonic on the electroporator at an output voltage of 2.2 kV. The mixture was quickly removed from the cuvette, transferred to 1 mL of yeast extract peptone (YEP) liquid broth (without antibiotics), and incubated for 3 h at 28 °C with shaking at 150 rpm. After incubation, the cells were harvested by centrifugation at 3000 rpm for 3 min, and the pellets were resuspended in 200 µL of YEP medium and grown on YEP plates containing 50 µg/mL rifampicin and 100 µg/mL spectinomycin. The YEP plates were incubated at 28 °C for two days. A single colony was used to inoculate each liquid culture (three constructs plus vector control) containing the above-mentioned antibiotics and incubated in a shaker (250 rpm) for two days at 28 °C. After two days of incubation, the cultures (100 µL) were then used to inoculate a 50 mL subculture of YEP medium containing the *Agrobacterium* strain and vector-specific antibiotics. The cultures were grown overnight at 28 °C with shaking. The cells were harvested by centrifugation at 5000 rpm for 10 min, and the pellets were resuspended in *Agrobacterium* infection medium (AIM, Appendix A) to yield an OD_600_ of 0.6. To improve plant transformation efficiency, 20 µL of acetosyringone stock (74 mM) was added to 40 mL of the AIM diluted cultures and kept on ice until internodal explants were treated with *Agrobacterium* to initiate transformation.

Internodal segments (5–10 mm in length) were excised from four-week-old in vitro plants, placed on callus induction medium (CIM, Appendix A) in 100 × 15 mm petri dishes (20 internodes per plate) and incubated in the dark for two days at 22 °C, followed by a light/dark photoperiod cycle of 16/8 h and 67 µmol m^−2^ s^−1^. For *Agrobacterium* infection, the internodal explants were transferred into previously prepared AIM and incubated for 20 min at room temperature with a shaker (50 rpm). Explants were transferred onto sterilized paper towels, blotted dry (to remove excess moisture), placed onto new CIM plates, and incubated for two days at 22 °C. Internodal segments were transferred into sterile falcon tubes, rinsed with 40 mL autoclaved double-distilled water (ddH_2_O) containing 250 mg/L cefotaxime for 5 min (3 times); blotted dry with sterilized paper towels; and transferred to CIM plates containing 500 mg/L carbenicillin, 250 mg/L cefotaxime, and 100 mg/L kanamycin (selection marker). The internode segments were transferred onto a fresh CIM every two weeks, and after four weeks, to shoot induction medium (SIM, Appendix A) supplemented with the same antibiotics. Once shoot primordia appeared and developed into plantlets at least 2 cm long, they were excised and transferred to Magenta^®^ boxes with root induction medium (RIM, Appendix A) supplemented with half-concentrations of the same antibiotics as SIM and incubated as before.

### 5.7. Genomic DNA Extraction and Genotype Analysis

To confirm transformation and the correct construct insertion into transgenic plants (*StFHT*, *StHCT*, and *St430*), gDNA was extracted from young leaves using 200 mM Tris HCl pH 7.5, 250 mM NaCl, 25 mM ethylenediaminetetraacetic acid (EDTA), and 0.5% sodium dodecyl sulfate (SDS). The extracted DNA was ethanol precipitated and dissolved in 50 µL of sterile water. PCR was performed with the gene- and vector-specific primers (Appendix A). The PCR conditions for the gene-specific primers were as follows: 94 °C for 4 min, 30 cycles of 94 °C for 30 s, 59 °C for 1.30 min (55 °C for vector-specific primers), 72 °C for 1.30 min, and a final extension time at 72 °C for 7 min.

### 5.8. Suberization Induction and Isolation of Potato Microtuber Wound Periderm

In preparation for the gene expression analysis and chemical analysis, potato microtubers were wounded by cutting them into quarters under sterile conditions in a laminar flow hood, put in Magenta^®^ boxes lined with stainless-steel mesh atop wet filter papers and incubated in the dark at 25 °C. For the metabolite analysis, minitubers were prepared as described [23]. After incubation, the suberized layers were collected by removing the top suberizing layer from cut surfaces with a razor, frozen with liquid nitrogen, and ground to a fine power with a pestle and a mortar. For RNA extraction, suberized tissues were collected aseptically after three days of wound healing and stored at −80 °C for RNA isolation or after seven days. Collected tissues was stored at −20 °C until used for the gene expression or chemical analysis. For analysis of the polar metabolites, suberized tissue was collected at 0, 12, 24, and 48 hpw. For the analysis of suberin monomers, suberized tissue was collected 7 days post-wounding.

### 5.9. RNA Isolation, cDNA Synthesis, and RT-qPCR Analysis

To confirm target gene knockdown in *StFHT*, *StHCT*, and *St430* plus the empty vector control, pK7GWIWG2D(II), genotypes, and total RNA was extracted from frozen, 3-day suberized tissues, as described by [35], with some modifications. Briefly, autoclaved extraction buffer composed of 2% hexadecyltrimethylammonium bromide (CTAB), 2% polyvinylpyrrolidinone K30 (PVP), 100 mM Tris HCl (pH 8.0), 25 mM EDTA, 2.0 M NaCl, 0.5 g/L spermidine, and 2% β-mercaptoethanol (added just before used) was warmed to 65 °C in a water bath. Approximately 20–100 mg of suberized tissue was ground in liquid nitrogen and mixed quickly (by inverting the tube) with 0.75 mL extraction buffer. An equal volume of chloroform: isoamyl alcohol (CHCl_3_: IAA) (24:1) was added and the samples vortexed for 5 s and centrifuged for 3 min at 10,000 rpm at room temperature in a microcentrifuge. The aqueous phase was collected and extracted twice more, with an equal volume of CHCl_3_: IAA (24:1). RNA was precipitated overnight at 4 °C by adding ¼ volume of 10 M LiCl to the aqueous phase and harvested by centrifugation at room temperature at 10,000 rpm for 30 min. After, the pellet was dissolved into 25 µL of SSTE buffer (1.0 M NaCl, 0.5% SDS, 10 mM Tris HCl (pH 8.0), and 1 mM EDTA) and precipitated again with 50 µL of 100% ethanol at −80 °C for 1 h. Finally, the pellet was collected by centrifugation at 10,000 rpm for 20 min, dried, and resuspended in 20 µL of diethylpyrocarbonate (DEPC)-treated water. To assess the purity of the RNA, the samples were evaluated using a NanoDrop™ One Microvolume UV–Vis Spectrophotometer (Thermo Scientific, Waltham, MA, USA) set for RNA determination. The purity of the RNA was evaluated following a spectrophotometer reading of A260/A280 >2.0 and A260/A30 ~2.0. RNA integrity was assessed using an Agilent 2100 Bioanalyzer (RIN ≥ 7). Next, the RNA samples (0.1–0.2 μg /50 μL) were treated with Turbo DNase to remove any contaminant DNA using the TurboDNA-free^™^ kit (Invitrogen, Waltham, MA, USA), and cDNA was synthesized using the Maxima Universal First Strand cDNA Synthesis Kit (Thermo Scientific) following the manufacturer’s protocols. Reverse transcription-quantitative PCR (RT-qPCR) analysis was performed to compare the gene expression between the wildtype, empty vector control, and the RNAi knockdown lines. The reaction was carried out using the synthesized cDNA, gene-specific primers (Appendix A), and PowerTrack™ SYBR Green Master Mix (Thermo Scientific) following the manufacture’s protocol. Primers for two endogenous controls, EF1-α and APRT [36], were used as a reference throughout the entire gene expression analyses, for data normalization (Appendix A). Primer efficiency was evaluated through a standard curve PCR reaction (in duplicate) using undiluted WT cDNA as the first point and performing a 1:10 dilution series (five points in total). To make sure there was no gDNA contamination, DNase-treated RNA was used as the negative control, and to identify any other type of contamination, water was used as the non-template control.

The expression level between the WT and five biological replicates of the empty vector were measured in quadruple using the target genes, specific primers (selected from the primer efficiency test). Once the expression level between the WT and the empty vector was determined, four biological replicates of the empty vector were used as the control to determine the knockdown level of the transgenic lines for each target gene. The reactions were measured in quadruple, and after the amplification cycles were completed, melted curves were generated to assess whether the RT-qPCR reactions produced a single, specific product. Normalized expression (∆∆Cq) between the target genes’ transgenic lines and the empty vector were determined using the Cq value generated in the Bio-Rad CFX Manager software (version 3.1) and the relative to zero options, as there were four biological replicates of the empty vector as the control group.

### 5.10. Polar Primary Metabolite Analysis

For polar metabolite analyses, frozen, ground periderm tissue (~10 mg) was extracted using a biphasic extraction protocol based on [37]. Briefly, with sample tubes kept on ice, 500 μL of cold (−20 °C) 50% *v*/*v* MeOH containing 0.025 mg/mL anthracene-9-COOH (C_15_H_10_O_2_; Exact Mass = 222.068085; [M + H]^+^ = 223.0685 *m*/*z*) [38] and 0.01 mg/mL ribitol [39] as internal standards was add to each sample, followed by the addition of 750 μL of cold (−20 °C) methyltertbutyl ether (M*t*BE). The tubes were thoroughly vortexed then placed on a rotating mixer in a cold room (5 °C) for 30 min. Samples were subsequently placed in a sonicating bath for 15 min in the cold room prior to their centrifugation in a bench top microcentrifuge for 10 min. To recover polar compounds, 100 μL aliquots of the MeOH-H_2_O phase were transferred to a clean microcentrifuge tube. For primary metabolites, recovered compounds were dried under nitrogen and prepared for GC-MS analysis by methoximation, followed by trimethylsilylation, according to [39]. Briefly, dried samples were reconstituted in 20 μL methoxamine HCl solution (20 mg/mL in pyridine) and incubated for 90 min at 70 °C. After cooling to room temperature, 80 μL MSTFA was added and the mixture incubated at 37 °C for 30 min. Derivatized samples were transferred to GC vials containing microvolume inserts and analyzed by GC-TOF-MS (see Section 5.12.1 below). For LC-TOF-MS analysis, recovered polar compounds were transferred to a screw cap chromatography vial and were directly analyzed by LC-MS in positive ion mode using a protocol based on [40] (see Section 5.12.2 below).

LC-MS raw data (mzdata format) were exported into XCMS Online (https://xcmsonline.scripps.edu/) for alignment. GC-MS raw data were exported to ChromaTOF Sync^®^ for alignment. The resulting aligned feature matrices were processed to remove features common in reagent blanks and normalized to the A9C (LCMS) and ribitol (GCMS) internal standards and tissue mass. MetaboAnalyst (https://www.metaboanalyst.ca/) was for untargeted feature analysis. For targeted metabolite analysis, features were identified in the aligned feature matrix by exact mass and retention time and co-chromatography with authentic standards where possible.

### 5.11. Soluble Waxes, Soluble Phenolic, and Aliphatic Suberin Analyses

Soluble lipids, including suberin-associated waxes, were extracted from frozen suberized periderm tissues (<100 mg) using a micro-Soxhlet extractor with a mixture of chloroform and methanol (CHCl_3_/CH_3_OH, 2:1; *v*/*v*) for 3.5 h (twice), followed by an overnight extraction with chloroform. The residual tissues were washed with acetone, air dried overnight in the fume hood, and stored at 4 °C for the insoluble suberin compounds extraction [41]. The chloroform and methanol extracts were pooled and dried on a rotary evaporator (Buchi, Switzerland) under vacuum at 40 °C; quantitatively transferred to 4 mL glass vials; and dried under a stream of nitrogen (N_2_) gas. The dried soluble residues were redissolved with chloroform and methanol (2:1; *v*/*v*) in a fixed volume/mass ratio (1 mL/10 mg) to normalize the extracts, and 100 µL aliquots were transferred to clean 4 mL glass vials and evaporated to dryness under nitrogen gas. The dried residues were then methylated (to yield methyl esters) by the addition of 3 M methanol/hydrochloric acid (500 µL) and heated at 80 °C for 2 h in a water bath. The samples were cooled to room temperature, and 1 mL of saturated NaCl was added to stop the reaction. Triacontane (10 µL of 1 mg/mL stock) was added as the internal standard, and the non-polar compounds were extracted three times with 1 mL hexane each time. The hexane extracts were pooled into a clean vial and dried under a stream of N_2_ gas. Finally, the samples were trimethylsilylated (TMS) by the addition of 15 µL each of pyridine and N,O-bis(trimethylsilyl)-trifluoroacetamide (BSTFA). After heating at 70 °C for 40 min in a water bath, the samples were cooled to room temperature and transferred to GC vials [41]. For calibration, a dilution series was prepared using ferulic acid standard, as well as fatty acids, fatty alcohols, and omega-hydroxy fatty acids (ω-OH FAs) (100 µg/mL–0.024 µg/mL). Samples were analyzed by GC-FID-MS (see Section 5.12.3 below).

#### 5.11.1. For Soluble Ferulates

Dried residues from 100 µL of the CHCl_3_/CH_3_OH (2:1; *v*/*v*) extracts (normalized to a fixed volume/mass ratio) were reconstituted in 40 µL CHCl_3_. Then, 10 µL of 0.1 mg/mL ergosterol (in ethanol) was added as the internal standard and the samples diluted to 100 µL with methanol. For calibration, a dilution series (100 mg/mL down to 0.78 mg/mL) was prepared using a C22-alkyl ferulate standard [42]. Standards were analyzed in triplicate. Alkyl ferulates were analyzed by LC-UV-MS in positive ion mode. Reconstituted samples (10 µL) were analyzed via LC-UV-MS (see Section 5.12.4 below).

#### 5.11.2. Insoluble Aliphatic Suberin Monomers

Insoluble aliphatic monomers were generated from chemically depolymerized extractive-free plant residues. Dried periderm tissues (2–3 mg) were trans-esterified to hydrolyze the poly(aliphatic) domain by incubation at 80 °C for 2 h with 500 µL of 3 M MeOH/HCl (Meyer et al. 2011). This process allows for the release of esterified aliphatics as methyl esters and alcohols. The released aliphatics were recovered, TMS-derivatized, and analyzed by GC-FID-MS (see Section 5.12.3 below).

#### 5.11.3. Total Poly(Phenolics)

Phenolic monomers were chemically released from the dried periderm tissues via microscale nitrobenzene oxidation (mNBO) according to [43], as modified by [44]. Briefly, dried periderm tissues (10 mg) were saponified with 1 M sodium hydroxide (1 mL) for 24 h at 37 °C, and this was repeated three times. The tissues were then washed with water (3 times), 80% methanol, and 100% acetone (one time with each solvent) and air dried in the fume hood. Subsequently, 2 M sodium hydroxide (NaOH, 300 µL) and nitrobenzene (15 µL) were added to the saponified tissues (1–2 mg) in a 5 mL glass ampoule. The ampoules were flame-sealed and incubated at 160 °C for 3 h. After cooling to room temperature, the ampoules were opened, and 5 µL of 20 mg/mL 3 ethoxy-4-hydroxybenzaldehyde was added as the internal standard. The samples were quantitatively transferred to 4 mL vials using saturated NaCl in water (about 2 mL), and the mixture was extracted twice with dichloromethane (1 mL) supernatant, as subsequently discarded. The aqueous phase was acidified (pH 2) with 1 M hydrochloric acid (HCl) and extracted again with 900 µL of hexanes (twice). The organic ethyl ether phases were collected, dried over anhydrous sodium sulfate (Na_2_SO_4_), and evaporated to dryness under a stream of nitrogen gas [43,44]. Finally, the samples were derivatized with 50 µL each of pyridine and BSTFA at 70 °C for 40 min and diluted with chloroform. Individual phenolic compounds were quantified using calibration curves derived from authentic standards (i.e., trimethylsilyl derivatives of p-hydroxybenzaldehyde, vanillin, vanillic acid, syringin, and syringic acid) in triplicate (μg/mL–μg/mL). Samples were immediately analyzed via GC-TOF-MS (see Section 5.12.5 below).

### 5.12. Chromatography-Mass Spectrometry

#### 5.12.1. GC-TOF-MS of Polar Metabolites

Samples were immediately analyzed via GC-MS on an Agilent 7890 A GC coupled with a LECO Pegasus BT time of flight MS. Liquid samples (1 μL) were injected in splitless mode onto a RESTEK Rxi-5ms Low bleed GC column (30 m, 250 μm internal diameter, and 0.25 μm film thickness) (Restek: Cat. 709-809-508) and eluted with the following oven temperature program: initial temperature at 50 °C held for 0.5 min followed by a temperature ramp at 20 °C min^−1^ to a final temperature of 325 °C and held at 325 °C for 5.75 min. Injector and transfer line temperatures were set to 275 °C. High purity helium was used as the carrier gas at a flow rate of 1 mL min^−1^. Data were acquired over a 50–500 *m*/*z* range after a 3-min delay to allow the solvent to clear the system.

#### 5.12.2. LC-TOF-MS of Polar Metabolites

LCMS separations were performed on an Agilent 1260 LC system (Agilent Technologies, Santa Clara, CA, USA) equipped with a C-18 column (Eclipse Plus RRHT, 3.0 × 100 mm, 1.8 μm; Agilent Technologies) by applying the following gradient, at a flow rate of 0.3 mL min^−1^, after a sample injection of 5 μL: 0 to 2 min, 95% A (0.1% *v*/*v* formic acid in Milli-Q H_2_O) in B (0.1% *v*/*v* formic acid in acetonitrile-H_2_O [9:1]); 12 min, 70% A, 30% B; 17 min, 100% B; hold at 100% B for 4 min. A 9-min post-run equilibration was completed at 100% A. ESI-TOF parameters: drying gas at 350 °C, 10 mL/min; nebulizer at 45 PSI; Vcap at 4000 V; Fragmentor at 130 V. Spectra were collected at 1.03/s (9729 transients/spectrum) in the 100–1700 *m*/*z* range. Reference mass solution (121.050873 *m*/*z* and 922.009798 *m*/*z*) was infused constantly via a second nebulizer at 15 psi.

#### 5.12.3. GC-FID-MS of Aliphatic Suberin Monomers

Separations were performed on a Varian CP-3800 Gas Chromatograph equipped with two detectors: the flame ionization detector (GC-FID) for quantification and the Varian MS220 ion trap mass spectrometer (GC-MS) for peak identification [41,45]. Briefly, the GC was equipped with two CP-Sil 5 CB low bleed MS columns (WCOT silica 30 m × 0.25 mm ID). One of the columns was in line with the FID and the other one was in line with the MS. The injector temperature was programmed to 250 °C, and the FID oven was programmed to 300 °C. The samples were injected to the columns (1 µL of samples for each column) in splitless mode, and the compounds were eluted using the following oven program: 70 °C for 2 min, increase to 200 °C (40 °C/min for 2 min), increase again to 300 °C (3 °C/min), and hold for 9.42 min, for a 50 min total run. The helium flow rate was set at 1 mL/min. Individual aliphatic compounds were quantified using calibration curves derived from authentic standards.

#### 5.12.4. LC-TOF-MS of Alkyl Ferulates

Alkyl ferulates were separated using an Agilent 1260 LC system (Agilent Technologies) equipped with a C-8 column (Eclipse Plus RRHT, 2.1 × 50 mm, 1.8 mm; Agilent), followed by an elution gradient of 70% A (0.1% formic acid in water) and 30% B (0.1% formic acid in acetonitrile) for 2 min, followed by a linear gradient to 100% B over 10 min. After 15 min at 100% B, the solvent conditions were brought back to 70% A/30% B and allowed to equilibrate for 12 min before the next sample was injected. The flow rate was set to 0.25 mL min^−1^. Ferulates were quantified by UV absorbance at 324 nm and identified by their exact mass ([M + H]^+^) using atmospheric pressure chemical ionization (APCI) in positive ion mode.

#### 5.12.5. GC-FID-MS of mNBO Products

The same GC-FID-MS system was used as detailed in GC-FID-MS Method 1. The injector temperature was programmed as 250 °C. The samples were injected to the column (1 µL, splitless mode), and the compounds were eluted using the following oven program: 140 °C for 4 min, increase to 240 °C (3 °C/min for 1 min), increase again to 310 °C (30 °C/min), and hold for 9.3 min, for a 50 min total run. The helium flow rate was set at 0.8 mL/min [44].

### 5.13. Periderm Permeability Measurements

Desiccation boxes were constructed out of magenta boxes filled with 2.5 cm depth of Dri-Rite^®^ and fitted with cardboard dividers. Wound periderm (n = 6 per line) and lenticel-free regions of the native periderm (n = 6 per line) were carefully sectioned and 1 cm diameter circular disks cut with a cork borer. Parafilm wrap and filter paper were used as positive and negative controls (n = 6), respectively. The periderm disks were fitted between the chromatography vial lid and a cored, Teflon-lined septa, leaving 12.6 mm^2^ of suberized tissue exposed. Lids were tightened onto vials containing 1.5 mL of ultra-pure water, until the periderms were firmly fixed in place with a water-tight seal. Vials were placed randomly into magenta boxes (6 vials per box) in an inverted position (periderm and lid construct facing down) to ensure the suberized tissue was in contact with the water. Weights of each vial were measured upon assembly and at regular intervals for up to eight days.

### 5.14. Statistical Analysis

All statistical analyses were performed using GraphPad Prism software R (version 10.2.1) with a preset probability value of 0.05 (*p* < 0.05) to identify significant differences. For comparison between wildtype and empty vector samples, pairwise Student’s *t*-tests were performed between four biological replicates of each. For qRT-PCR, suberin monomer, and targeted metabolite analyses, a one-way ANOVA, followed by Holm–Šídák’s post hoc test was used. For untargeted metabolite analyses, PLS-DA was performed in MetaboAnalyst.

## Figures and Tables

**Figure 1 plants-13-02995-f001:**
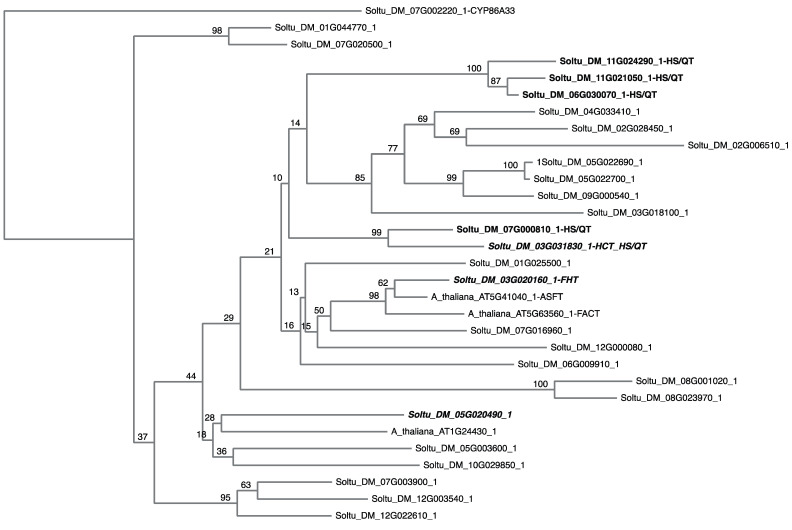
Phylogeny of wound-induced potato HXXXD/BAHD genes. The amino acid sequences for 28 HXXXD/BAHD acyltransferases induced by wounding were aligned and used to create a neighbor-joining phylogenetic tree (1000 bootstraps) using MAFFT (mafft.cbrc.jp). Amino acid sequences for three *Arabidopsis* HXXXD/BAHD genes (At5g41040, At5g63560, and At1g24430) were also included. StCYP86A33 (Soltu.DM.07G002220.1) was used as an outgroup to root the tree. The rooted phylogenetic tree was rendered using Phylo.IO (http://phylo.io/).

**Figure 2 plants-13-02995-f002:**
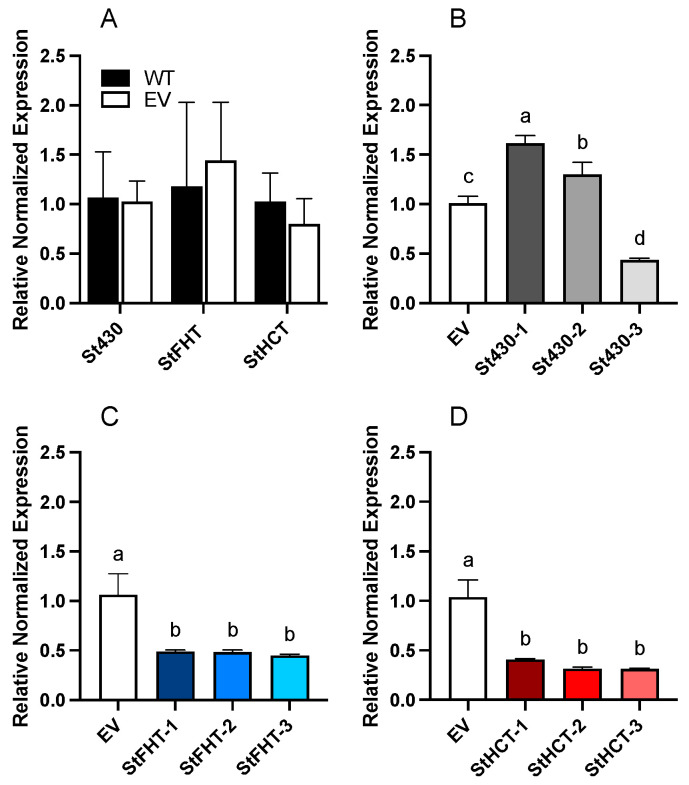
Target gene expression in potato RNAi-knockdown lines. (**A**) Expression levels of *St430*, *StFHT*, and *StHCT* in wildtype (WT) and empty vector (EV) lines of potatoes, 72 hpw. (**B**) Expression of *St430* in EV and three putative RNAi knockdown lines, 72 hpw. (**C**) Expression of *StFHT* in EV and three putative RNAi knockdown lines, 72 hpw. (**D**) Expression of *StHCT* in EV and three putative RNAi knockdown lines, 72 hpw. Data within each panel were analyzed separately by one-way ANOVA, followed by Holm–Šídák’s multiple comparisons test. Different letters above bars indicate significant differences (*p* = 0.05).

**Figure 3 plants-13-02995-f003:**
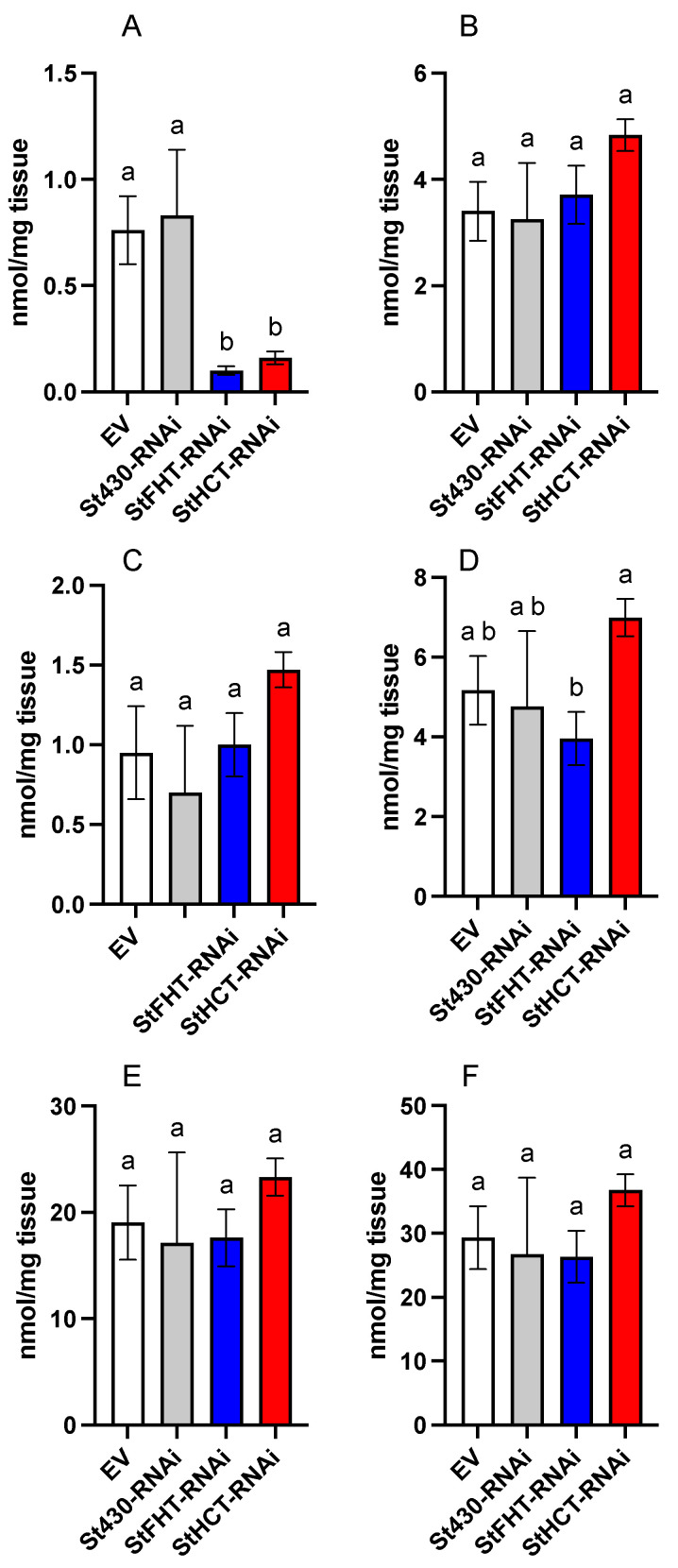
Aliphatic suberin analysis of potato wound periderm. Aliphatic suberin monomers were released from wound periderm collected 168 hpw following MeOH-HCl transesterification. Derivatized monomers were analyzed by GC and pooled into chemical classes: (**A**) esterified ferulic acid, (**B**) fatty acids, (**C**) 1-alkanols, (**D**) ω-hydroxy fatty acids, (**E**) α,ω -dioic acids, and (**F**) total aliphatic suberin monomers. There were no statistical differences in the number of aliphatic monomers between independent lines within EV, StFHT-RNAi, and StHCT-RNAi. Data for five independent EV lines were pooled (*n* = 5). For the StFHT-RNAi and StHCT-RNAi lines, five replicates from each of three independent lines were pooled (*n* = 15). For the St430-RNAi lines, data from three independent replicates were pooled (*n* = 3). Within each class of monomers, the data were analyzed by one-way ANOVA, followed by Holm–Šídák’s multiple comparisons test. Different letters above bars indicate significant differences (*p* = 0.05).

**Figure 4 plants-13-02995-f004:**
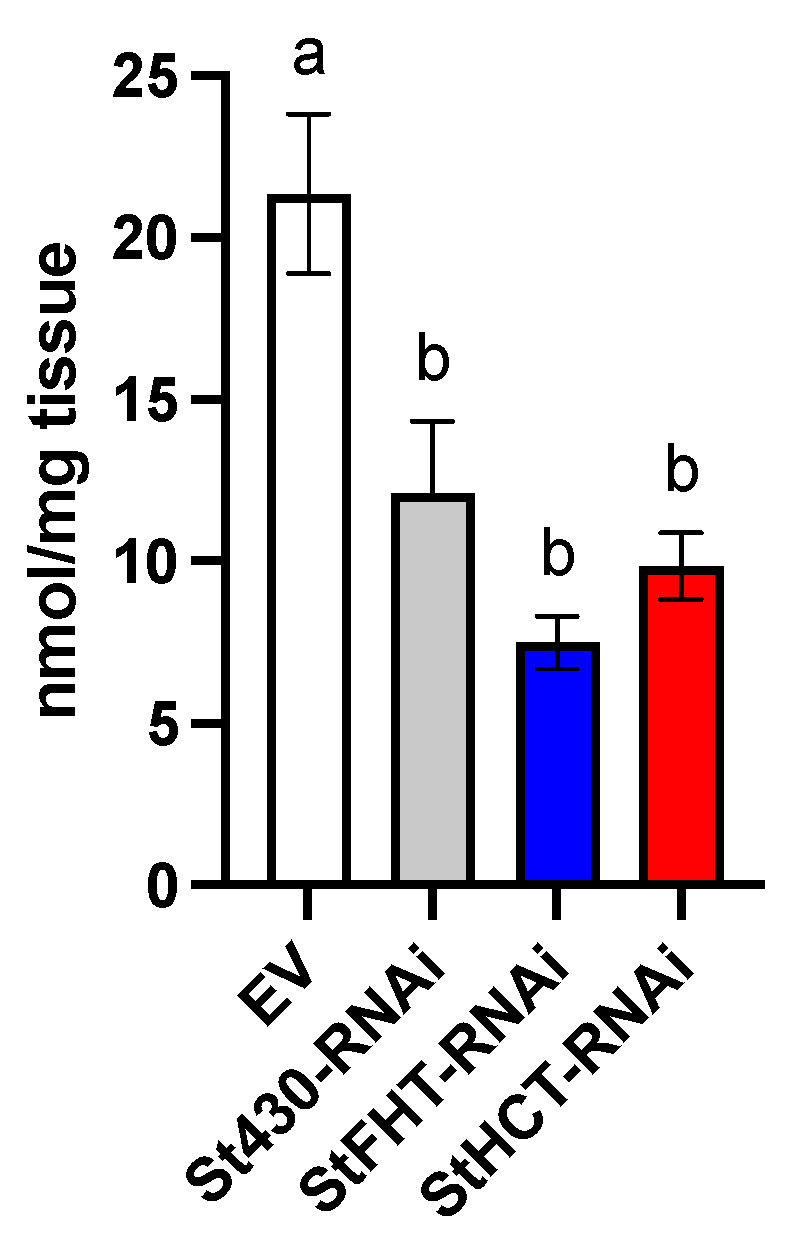
Soluble alkyl ferulates in potato RNAi-knockdown line wound periderm. Ferulate esters were extracted from wound periderm collected 168 hpw and analyzed by LCMS. There were no statistical differences in the number of alkyl ferulates between independent lines within EV, StFHT-RNAi, and StHCT-RNAi. Data for five independent EV lines were pooled (*n* = 5). For the StFHT-RNAi and StHCT-RNAi lines, five replicates from each of three independent lines were pooled (*n* = 15). For the St430-RNAi lines, data from three independent replicates were pooled (*n* = 3). Data were analyzed by one-way ANOVA, followed by Holm–Šídák’s multiple comparisons test. Different letters above bars indicate significant differences (*p* = 0.05).

**Figure 5 plants-13-02995-f005:**
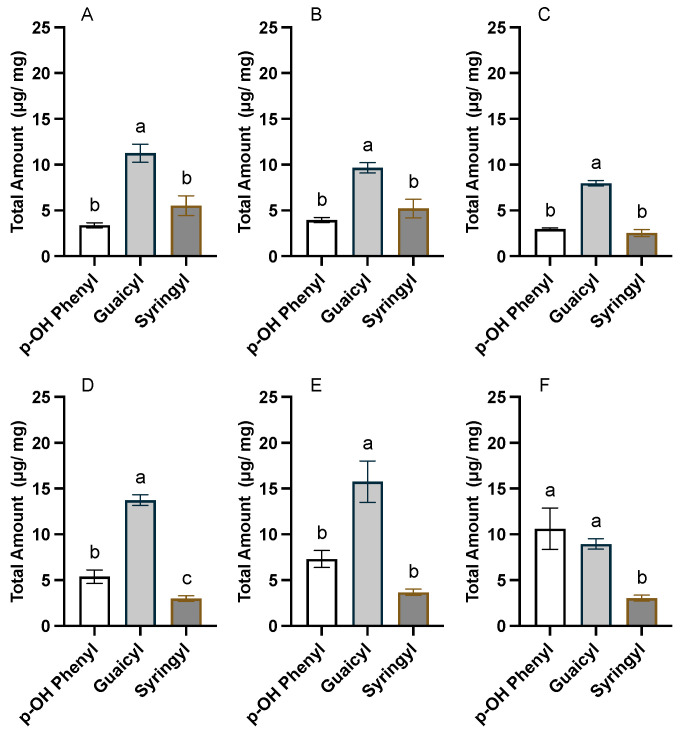
Composition of phenolic suberin from potato native and wound periderm. mNBO was applied to native (**A**–**C**) and wound (168 hpw) (**D**–**F**) periderm tissue collected from EV (**A**,**D**), StFHT-RNAi (**B**,**E**), and StHCT-RNAi (**C**,**F**) lines. There were no statistical differences in the number of phenolic monomers between independent lines within EV, StFHT-RNAi, and StHCT-RNAi. Native and wound periderm for five independent EV lines were pooled and analyzed. For StFHT-RNAi lines, *n* = 7 for wound periderm and *n* = 9 for native periderm, while, for StHCT-RNAi lines, *n* = 14 for native and wound periderm. Data were analyzed by one-way ANOVA, followed by Holm–Šídák’s multiple comparisons test. Different letters above bars indicate significant differences (*p* = 0.05).

**Figure 6 plants-13-02995-f006:**
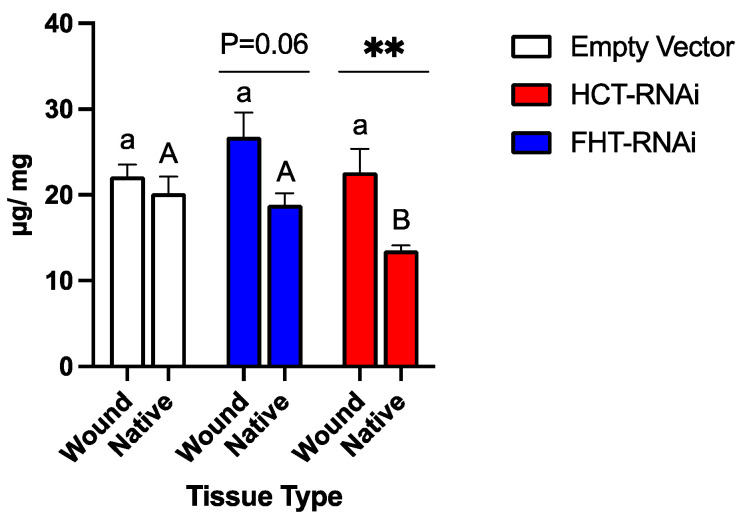
Total phenolic suberin in potato native and wound periderm. mNBO was applied to native and wound (168 hpw) periderm tissue collected from EV, StFHT-RNAi knockdown, and StHCT-RNAi lines. There were no statistical differences in the amount of phenolic monomers between independent lines within EV, StFHT-RNAi, and StHCT-RNAi. Native and wound periderm for five independent EV lines were pooled and analyzed. For StFHT-RNAi lines, *n* = 7 for wound periderm and *n* = 9 for native periderm, while, for StHCT-RNAi lines, *n* = 14 for native and wound periderm. *p*-Hydroxyphenyl-, guaiacyl-, and syringyl-derived moieties were summed for each replicate. Different lowercase letters signify significant differences between wound periderm data across genotypes, while upper case letters denote significant differences between native periderm data across genotypes (*p* < 0.05). Within-genotype comparisons between native and wound periderm are denoted by a line above pairs of bars (** = *p* < 0.01).

**Figure 7 plants-13-02995-f007:**
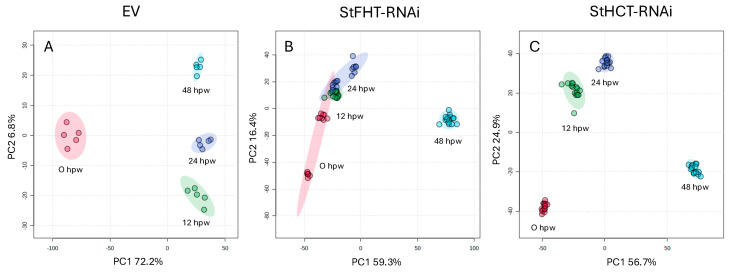
Changes in potato polar metabolite profiles post-wounding. Soluble polar metabolites were extracted from EV (**A**), StFHT-RNAi (**B**), and StHCT-RNAi (**C**) lines and analyzed by LCMS. Normalized data were imported into MetaboAnalyst^®^ and used to generate PLS-DA score plots. Data from five independent EV lines and five replicates from each of three StFHT- and StHCT-RNAi lines were pooled separately for each analysis. Each data point represents a unique polar metabolite profile from an independent replicate.

**Figure 8 plants-13-02995-f008:**
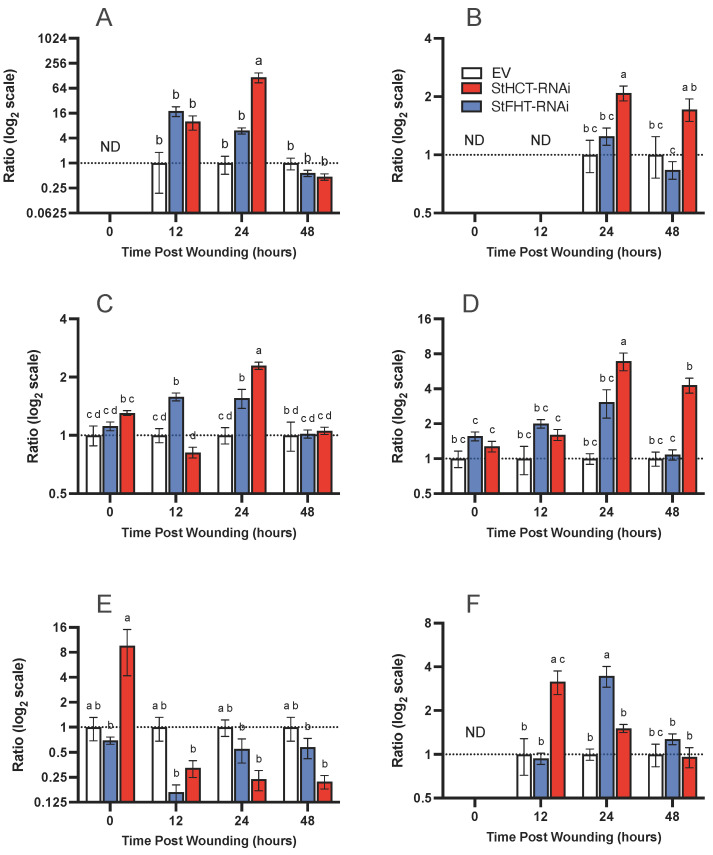
Phenolic metabolite accumulation in potato lines post-wounding, Targeted analysis of (**A**) p-coumaroyl tyramine, (**B**) caffeic acid, (**C**) chlorogenic acid, (**D**) caffeoyl putrescine, (**E**) feruloyl putrescine, and (**F**) feruloyl tyramine was used to quantify soluble polar metabolites extracted from wound-healing potato tubers from EV, StFHT-, and St-HCT-RNAi lines. There were no statistical differences in the number of soluble phenolics between independent lines within EV, StFHT-RNAi, and StHCT-RNAi. Native and wound periderm for five independent EV lines were pooled and analyzed. For StFHT-RNAi and StHCT-RNAi lines, five replicates from each of three independent lines were pooled (*n* = 15). Relative data (normalized to EV set at 1; dashed lines) are plotted on a log_2_ scale. Data were analyzed using a two-way ANOVA, followed by a Holm–Šídák’s multiple comparisons test. Different letters above bars indicate significant differences (*p* = 0.05). ND = not detected.

**Figure 9 plants-13-02995-f009:**
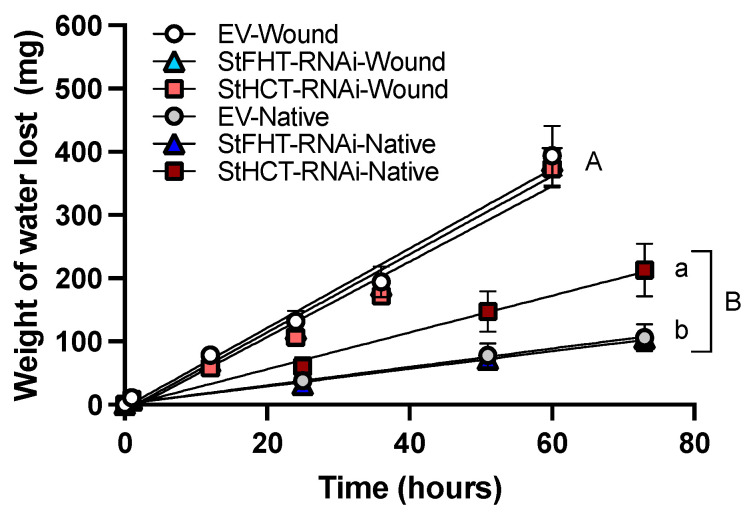
Potato periderm permeability. Permeability of potato periderm tissue was measured by securing an intact piece of native or wound (168 hpw) periderm across an open top vial containing 1 mL ultrapure water. Vials were suspended upside down (to ensure water contact with the tissue) in containers lined with Dri-Rite^®^. Water loss was recorded as weight loss over time. No weight loss was observed for vials lined with Parafilm^®^ (negative control), while nearly all water was lost through cellulose paper (positive) controls within 24 h. Slopes were analyzed using a one-way ANOVA, followed by a Holm–Šídák’s multiple comparisons test. Different uppercase letters indicate significant differences (*p* = 0.05) between wound and native periderm samples. Lowercase letters denote significant differences (*p* = 0.05) between wound periderm samples.

**Figure 10 plants-13-02995-f010:**
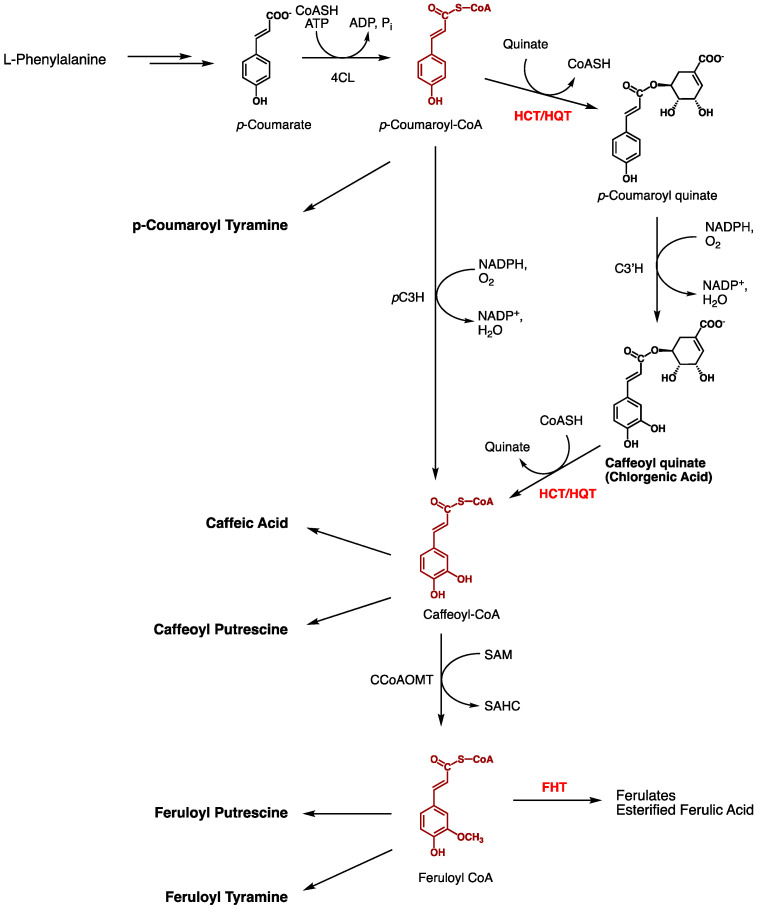
Phenylpropanoid metabolism. The origin of suberin-derived phenylpropanoids is shown, highlighting the placement of StHCT and StFHT. StFHT catalyzes the transfer of ferulic acid moieties from feruloyl-CoA to acyl acceptors (i.e., ferulates) and suberin aliphatic monomers (i.e., yielding esterified ferulic acid). In contrast, StHCT is hypothesized to transfer *p*-coumaric acid moieties from p-coumaroyl-CoA to quinate (or shikimate) acceptors prior to their hydroxylation to form caffeoyl quinate(shikimate) via *p*-coumaroyl quinate-3′-hydroxylase (C3′H). The caffeoyl moiety from caffeoyl quinate is then transferred to CoASH by the reverse StHCT reaction. The CoA derivatives of *p*-coumaric, caffeic, and ferulic acids are hypothesized to be precursors to the putrescine and tyramine amides that accumulate in potato tubers post-wounding. Names of compounds found to accumulate in StFHT- and StHCT-RNAi lines are in boldface type. Abbreviations: 4CL, 4-coumaroyl-CoA ligase; *p*C3H, *p*-coumarate-3-hydroxylase; HCT/HQT, hydroxycinnamoyl-CoA shikimate/quinate hydroxycinnamoyl transferase; C3′H, *p*-coumaroyl quinate-3′-hydroxylase; CCoAOMT, caffeoyl-CoA O-methyl transferase; FHT, fatty ω-hydroxyacid/fatty alcohol hydroxycinnamoyl transferase.

## Data Availability

The datasets generated during and/or analyzed during the current study are available from the corresponding author upon reasonable request.

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
