# Peer review of "Altered Metabolism in Knockdown Lines of Two HXXXD/BAHD Acyltransferases During Wound Healing in Potato Tubers"

_plants, 2024, doi:10.3390/plants13212995_

Round 1
Reviewer 1 Report
Comments and Suggestions for Authors
I checked your manuscript and described comments below.
The size of potato tubers has a large impact on their commercial value.
This paper provides an excellent analysis of metabolic changes in HXXXD/BAHD 2 acyltransferase knockdown lines.
I think it would be a good idea to consider the following points.
1. It would be better if you had photos of RNAi-knockdown potato and normal potato.
2. I think it would be better to unify the vertical scales of the graphs in Figures 2 and 3.
3. Mafft is the software that does the alignment. Neighbour-joining phylogenetic tree (1000 bootstrap) is done with another software. It is recommended to check the software used to calculate the phylogenetic tree.
4. Ref17 is a master's thesis. If possible, it would be better to have it published in an international academic journal. Or, since it is a co-author's paper, it may be fine to include it in this paper.
I don't think this paper has major problems and grammatical problems.
Author Response
Comment 1: It would be better if you had photos of RNAi-knockdown potato and normal potato.
Response 1: There is no visible phenotype for any of the RNAi knockdown potato lines, making photos uninformative. Instead of photos, we have added the statement “There was no visible phenotype for any of the RNAi knockdown potato lines
Comment 2: I think it would be better to unify the vertical scales of the graphs in Figures 2 and 3.
Response 2: We have unified the y-axes for figure 2; however, in figure 3 the axes vary because the amounts of different monomer classes vary considerably. If we unify these axes, some of the monomer amounts will become too low to see on the graphs. We have left these in their original format.
Comment 3: Mafft is the software that does the alignment. Neighbour-joining phylogenetic tree (1000 bootstrap) is done with another software. It is recommended to check the software used to calculate the phylogenetic tree.
Response 3: In the online version of MAFFT, there is an option to create a phylogenetic tree based on the alignment done within the software. This is what was used to generate Figure 1. As noted in both the figure caption and M&M, the rooted phylogenetic tree was rendered using Phylo.IO (http://phylo.io/).
Comment 4: Ref17 is a master's thesis. If possible, it would be better to have it published in an international academic journal. Or, since it is a co-author's paper, it may be fine to include it in this paper.
Response 4: Some of the data in the manuscript comes from the MSc thesis (ref 17) of co-author Indira Queralta-Castillo. It is included as a reference where information not included in the manuscript, but is in the thesis, is relevant.
Reviewer 2 Report
Comments and Suggestions for Authors
Article «Altered metabolism in knockdown lines of two HXXXD/BAHD acyltransferases during wound healing in potato tubers» of the authors Jessica L. Sinka, Indira Queralta-Castillo, Lorena S. Yeung, Isabel Molina, Sangeeta Dhaubhadel and Mark A. Bernards is devoted to the suberin synthesis investigation in the wound-induced suberin phenotype of Solanim tuberosum
The article is well presented, the results are clearly introduced, the current results are discussed and compared with the results of other authors, and the conclusions correspond to the obtained results.
Comments
- Please complete the introduction with a reason for choosing the researching object. Why is it important to investigate the synthesis of suberin in potato tubers?
- Include in the text of the introduction the practical significance of the current work
It is necessary to emphasize the practical importance for the crop breeding development or for other scientific research areas of the current research
- All abbreviations must be deciphered at the first mention:
Line 94 – WT
- Section 2.2. Selection of RNAi knockdown Lines
When describing Figure 2, there is no mention of the B-D subfigure s in the text.
- Section 2.6. Targeted metabolite analysis reveals a potential role for StHCT early in phenylpropanoid metabolism
- Line 246-247
Do the authors mean Figures 5 C and F?
If so, then I see a different picture with regard to p-hydroxyphenyl.
After wounding, the amount of p-hydroxyphenyl increases significantly (Fig.5 F).
The Sentence on lines 246-247 is not entirely clear. Please rephrase it more clearly.
- Line 325 «…loss from wound periderm from any genotype.» the link to Figure 9A is missing
- Line 327 «(Figure 9).» - Figure 9 B?
- Line 391 «…wound phenolic wound suberin..» the second mention of wound should be deleted.
In the section "Materials and methods" it is necessary to check the correspondence of the numbering of the cited subsections: line 723, 731, 738, 760
Author Response
Comment 1: Please complete the introduction with a reason for choosing the researching object. Why is it important to investigate the synthesis of suberin in potato tubers?
Include in the text of the introduction the practical significance of the current work
It is necessary to emphasize the practical importance for the crop breeding development or for other scientific research areas of the current research
Response 1: We have added text to the introduction (new lines 45-47 and 108-110) that provides more rationale for the work.
Comment 2: All abbreviations must be deciphered at the first mention:
Line 94 – WT
Response 2:The WT abbreviation has been defined (new line 97)
Comment 3: Section 2.2. Selection of RNAi knockdown Lines
When describing Figure 2, there is no mention of the B-D subfigure s in the text.
Reponse 3: Figure 2 subsections have now been mentioned in the text (new lines 159-161)
Comment 4: Section 2.6. Targeted metabolite analysis reveals a potential role for StHCT early in phenylpropanoid metabolism
Line 246-247
Do the authors mean Figures 5 C and F?
If so, then I see a different picture with regard to p-hydroxyphenyl.
After wounding, the amount of p-hydroxyphenyl increases significantly (Fig.5 F).
The Sentence on lines 246-247 is not entirely clear. Please rephrase it more clearly.
Response 4: We have added text (new lines 254-256) to make clear we are referring to changes in the p-hydroxyphenyl moieties in the wound periderm
Comment 5: Line 325 «…loss from wound periderm from any genotype.» the link to Figure 9A is missing
Line 327 «(Figure 9).» - Figure 9 B?
Response 5: There are no sub panels for Figure 9. The A and B in the figures denotes the statistical difference in permeability measures between wound and native periderm. Lower case a and b refer to statistical differences in native periderm permeability between genotypes.
Comment 6: Line 391 «…wound phenolic wound suberin..» the second mention of wound should be deleted.
Response 6: The second mention of “wound” has been deleted (new line 399).
Comment 7: In the section "Materials and methods" it is necessary to check the correspondence of the numbering of the cited subsections: line 723, 731, 738, 760
Response 7: The subsection citations for chromatography have been corrected.
Reviewer 3 Report
Comments and Suggestions for Authors
The paper “Altered metabolism in knockdown lines of two HXXXD/BAHD acyltransferases during wound healing in potato tubers” will be a good addition to the wound healing-related studies in potato tubers. However, I have concerns over paper, which need further explanation. The major concerns are provided below:
- Why was the HXXXD/BAHD family specifically selected for the study?
- The author directly selected some genes and generated RNAi lines. Why the authors didn’t perform qPCR for all the HXXXD/BAHD genes under specific wound healing conditions and just chose 1-2 important genes for study. In this case, there will be no need to generate so many RNAi lines.
- I would also suggest making a graphical abstract for clarification of the results.
- I would suggest adding some plant phenotype pictures to the paper.
The other minor comments are provided below for improvement.
Title
The title can be changed to “Metabolic Alterations in Two Potato HXXXD/BAHD Acyltransferase Knockdown Lines During Wound Healing.”
Abstract
Line 14: Before this line, add one line for the wound healing importance in potato.
Line 18: Change RNAseq to RNA-seq.
Introduction
There must be a paragraph in the introduction that explains the role of wound healing in potatoes. And should be placed at the beginning of the introduction.
Results
Figure 1: I would suggest converting the phylogenetic tree to a more beautiful circular form, where it should be clearly indicated, with different symbols or another method, which genes belong to potato and which genes belong to Arabidopsis. Even the phylogenetic tree in portrait orientation will be more appealing and easy to read.
Lines 150-151: The lines need to be modified for better clarity. Where the empty vector line can be combined with the first line.
Figure 2A: Provide significance letters above bars.
Line 208: I would suggest adding some phenotype pictures in the paper.
Conclusion
The conclusion section needs to be modified for better clarification of the authors results.
Author Response
No review yet.